# Theoretically Provable Spiking Neural Networks

**Shao-Qun Zhang**    **Zhi-Hua Zhou**$^*$

National Key Laboratory for Novel Software Technology
Nanjing University, Nanjing 210093, China
`{zhangsq,zhouzh}@lamda.nju.edu.cn`

## Abstract

Spiking neural networks have attracted increasing attention in recent years due to their potential of handling time-dependent data. Many algorithms and techniques have been developed; however, theoretical understandings of many aspects of spiking neural networks are far from clear. A recent work [44] disclosed that typical spiking neural networks could hardly work on spatio-temporal data due to their bifurcation dynamics and suggested that the *self-connection* structure has to be added. In this paper, we theoretically investigate the approximation ability and computational efficiency of spiking neural networks with self connections, and show that the self-connection structure enables spiking neural networks to approximate discrete dynamical systems using a polynomial number of parameters within polynomial time complexities. Our theoretical results may shed some insight for the future studies of spiking neural networks.

## 1 Introduction

The past decades have witnessed an increasing interest in spiking neural networks (SNNs) due to their great potential of modeling time-dependent data [28, 32, 37]. The fundamental units of SNNs are usually formulated as the combination of an integrated process (e.g., some first-order parabolic equations) and a step firing function. There has been significant progress on computational and implementation techniques for SNNs in computer vision [13, 33], speech recognition [30, 39], reinforcement learning [11, 38], few-short learning [16, 22, 25], etc. However, theoretical understandings of many aspects of SNNs, such as the approximation ability and computational efficiency on spatio-temporal systems, are far from clear.

Some researchers [19, 20, 21, 31] focused on the approximation universality of SNNs, in which some typical SNNs can simulate the standard computational models such as Turing machines, random access machines, threshold circuits, sequence-to-sequence mapping, etc. There are also efforts on the computational efficiency of SNNs for some specific issues, such as the convergence in the limit results and computational complexity of SNNs for the sparse coding problem [34, 35] and temporal quadratic programming [5, 8], respectively.

Amazingly, a recent study [44] theoretically proved that, contrary to previous beliefs, typical SNNs can hardly work well on spatio-temporal data, because they in nature are bifurcation dynamical systems with fixed eigenvalues in which many patterns inherently cannot be learned. They also suggested that adding *self-connection* structure can enhance the representation ability of SNNs on spatio-temporal systems that fully connect the spiking neurons in the same layer and solves adaptive eigenvalues of discrete dynamical systems.

---

$^*$Zhi-Hua Zhou is the corresponding author.

36th Conference on Neural Information Processing Systems (NeurIPS 2022).

In this paper, we theoretically investigate the approximation ability and computational efficiency of the self-connection spike neural networks (scSNNs). Our theoretical results show that equipped with self connections, scSNNs can approximate discrete dynamical systems using polynomial number of parameters within polynomial time complexities. Our main contributions are summarized as follows:

- We prove that the proposed scSNNs are universal approximators in Theorem 1.
- As for spatial approximation, we prove that a broad range of radial functions can be well approximated by scSNNs with polynomial spiking neurons in Theorem 2.
- As for temporal approximation, we prove that multivariate spike flows can be approximated by scSNNs within polynomial time in Theorem 3 and verify this conclusion in simulation experiments.

The rest of this paper is organized as follows. Section 2 introduces notations. Section 3 presents the scSNNs and some related concepts. Section 4 provides three key theorems to show the universal approximation ability and (parameters and time) complexity of scSNNs for approximating the discrete dynamical system. Section 5 develops in-depth discussions of the effects led by self-connection. Section 6 concludes this work.

## 2 Notations

Here, we provide some useful notations, which are detailed in Appendix. Let $[N] = \{1, 2, \ldots, N\}$ be an integer set for $N \in \mathbb{N}^+$, and $|\cdot|_\#$ denotes the number of elements in a collection, e.g., $|[N]|_\# = N$. Let $i = \sqrt{-1}$ be the imaginary unit, and $\boldsymbol{x} \preccurlyeq 0$ means that every element $x_i \leq 0$ for any $i$. Let the sphere $\mathcal{S}(r)$ and globe $\mathcal{B}(r)$ be $\mathcal{S}(r) = \{\boldsymbol{x} \mid \|\boldsymbol{x}\| = r\}$ and $\mathcal{B}(r) = \{\boldsymbol{x} \mid \|\boldsymbol{x}\| \leq r\}$ for any $r \in \mathbb{R}$, respectively. Given a function $g(n)$, we denote by $h_1(n) = \Theta(g(n))$ if there exist positive constants $c_1, c_2$ and $n_0$ such that $c_1 g(n) \leq h_1(n) \leq c_2 g(n)$ for every $n \geq n_0$; $h_2(n) = \mathcal{O}(g(n))$ if there exist positive constants $c$ and $n_0$ such that $h_2(n) \leq cg(n)$ for every $n \geq n_0$; $h_3(n) = \Omega(g(n))$ if there exist positive constants $c$ and $n_0$ such that $h_3(n) \geq cg(n)$ for every $n \geq n_0$; $h_4(n) = o(g(n))$ if there exists positive constant $n_0$ such that $h_4(n) < cg(n)$ for every $c > 0$ and $n \geq n_0$.

Let $\mathcal{C}(K, \mathbb{R})$ be the set of all scalar functions $f : K \to \mathbb{R}$ continuous on $K \subset \mathbb{R}^n$. Given $\boldsymbol{\alpha} = (\alpha_1, \alpha_2, \ldots, \alpha_m)^\top \in \mathbb{N}^m$, we define

$$D^{\boldsymbol{\alpha}} f(\boldsymbol{x}) = \frac{\partial^{\alpha_1}}{\partial x^{\alpha_1}} \frac{\partial^{\alpha_2}}{\partial x^{\alpha_2}} \cdots \frac{\partial^{\alpha_m}}{\partial x^{\alpha_m}} f(\boldsymbol{x}) \,,$$

where $\boldsymbol{x} = (x_1, x_2, \ldots, x_n) \in K$. Further, we define

$$\mathcal{C}^l(K, \mathbb{R}) = \{f \mid f \in \mathcal{C}(K, \mathbb{R}) \text{ and } D^r f \in \mathcal{C}(K, \mathbb{R}), \text{ for } r \in [l]\} \,.$$

For $1 \leq p < \infty$, we define

$$\mathcal{L}^p(K, \mathbb{R}) = \left\{ f \;\middle|\; f \in \mathcal{C}(K, \mathbb{R}) \text{ and } \|f\|_{p,K} \triangleq \left( \int_K |f(\boldsymbol{x})|^p \, \mathrm{d}\boldsymbol{x} \right)^{1/p} < \infty \right\} \,.$$

This work considers the Sobolev space $\mathcal{W}_\mu^{l,p}(K, \mathbb{R})$, defined as the collection of all functions $f \in \mathcal{C}^l(K, \mathbb{R})$ and $D^r f \in \mathcal{L}^p(K, \mathbb{R})$ for all $|\boldsymbol{\alpha}| \in [l]$, that is,

$$\|D^{\boldsymbol{\alpha}} f\|_{p,K} = \left( \int_K |D^{\boldsymbol{\alpha}} f(\boldsymbol{x})|^p \, \mathrm{d}\boldsymbol{x} \right)^{1/p} < \infty \,.$$

This paper employs $\mathbf{E}_n$ to denote the $n \times n$ unit matrix and $\det(\cdot)$ to indicate the determinant operation on the matrix. Two $n$-by-$n$ matrices $\mathbf{A}$ and $\mathbf{B}$ are called *similar*, denoted as $\mathbf{A} \sim \mathbf{B}$, if there exists an invertible $n \times n$ matrix $\mathbf{P}$ such that $\mathbf{B} = \mathbf{P}^{-1} \mathbf{A} \mathbf{P}$. The general linear group over field $\mathbb{F}$, denoted as $\mathbf{GL}(n, \mathbb{F})$, is the set of $n \times n$ invertible matrices with entries in $\mathbb{F}$. Especially, we define that a special linear group $\mathbf{SL}(n, \mathbb{F})$ is the subgroup of $\mathbf{GL}(n, \mathbb{F})$ and consists of matrices with determinant 1. For any field $\mathbb{F}$, the $n \times n$ orthogonal matrices form the following subgroup

$$\mathbf{O}(n, \mathbb{F}) = \{\mathbf{P} \in \mathbf{GL}(n, \mathbb{F}) \mid \mathbf{P}^\top \mathbf{P} = \mathbf{P} \mathbf{P}^\top = \mathbf{E}_n\}$$

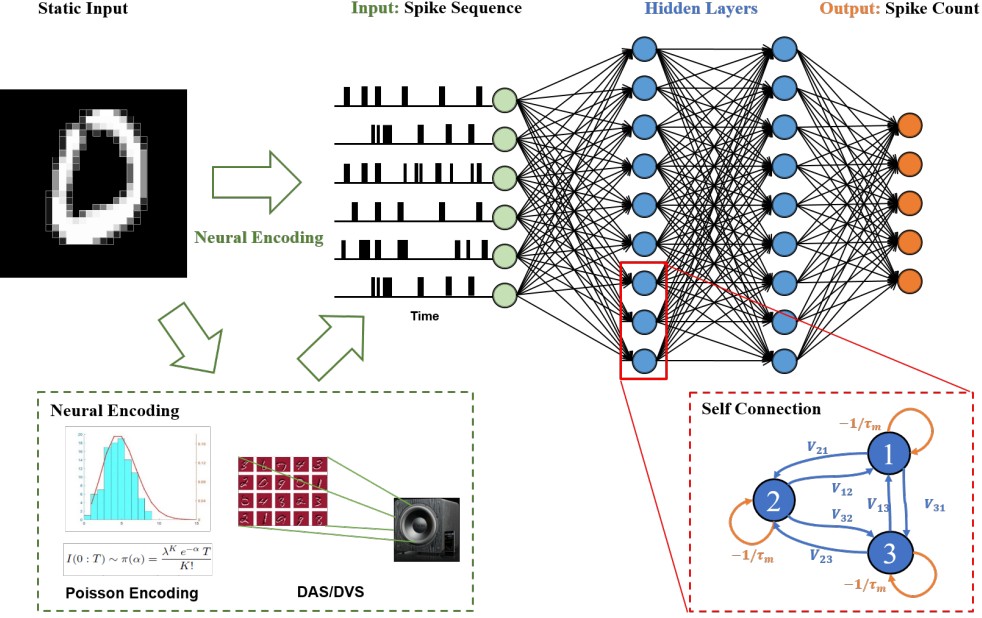

Figure 1: Illustrations of scSNNs and Self Connections.

of the general linear group $\mathbf{GL}(n, \mathbb{F})$. Similarly, we have the special orthogonal group, denoted as $\mathbf{SO}(n, \mathbb{F})$, which consists of all orthogonal matrices of determinant 1 and is a normal subgroup of $\mathbf{O}(n, \mathbb{F})$. This group is also called the rotation group, generalizing that linearly transforms geometries while holding the surface orientation.

Let $\phi^2(\boldsymbol{x})$ be the density function of some probability measure $\mu$, which satisfies
$$\int_{\boldsymbol{x} \in \mathbb{R}^m} \phi^2(\boldsymbol{x}) \, \mathrm{d}\boldsymbol{x} = \int_{\boldsymbol{x} \in \mathcal{B}(1)} 1 \, \mathrm{d}\boldsymbol{x} = 1 \ .$$
For continuous functions $f$ and $g$, we have the following equalities under Fourier transform
$$\|\widehat{f\phi} - \widehat{g\phi}\|_{L_2(\mu)} = \|f\phi - g\phi\|_{L_2(\mu)} \quad \text{and} \quad \widehat{f\phi} = \hat{f} * \hat{\phi} \ ,$$
for the convolution operator $*$.

## 3 Self-connection Spiking Neural Networks

Let $n$ be the number of spiking neurons, $\boldsymbol{u}(t) \in \mathbb{R}^n$ and $\boldsymbol{s}(t) \in \mathbb{R}^n$ denote the vectors of membrane potentials and spikes in which $\boldsymbol{u}_i(t)$ and $\boldsymbol{s}_i(t)$ are the membrane potential and spikes of neuron $i \in [n]$ at time $t \geq 0$, respectively. Inheriting the recognition in [44], we here consider the self-connection SNN (scSNN) as follows:
$$\frac{\mathrm{d}\boldsymbol{u}_i(t)}{\mathrm{d}t} = -\frac{1}{\tau_m}\boldsymbol{u}_i(t) + \sum_{j \in [n]} \mathbf{V}_{ij}\boldsymbol{s}_j(t) + \sum_{k \in [m]} \mathbf{W}_{ik}\mathbf{I}_k(t) \ , \tag{1}$$

where $\tau_m$ is a positive-valued hyper-parameter with respect to membrane time, $\mathbf{I}_k(t)$ indicates the signal from input channel $k \in [m]$ at time $t$, $\mathbf{V} \in \mathbb{R}^{n \times n}$ and $\mathbf{W} \in \mathbb{R}^{n \times m}$ denote the self-connection and connection weights matrices, respectively. Hence, $\sum_{k \in [m]} \mathbf{W}_{ik}\mathbf{I}_k(t)$ indicates the signal received by neuron $i$ at time $t$, and $\sum_{j \in [n]} \mathbf{V}_{ij}\boldsymbol{s}_j(t)$ denotes the effect on the membrane potential of neuron $i$ when neuron $j$ fires a spike, as illustrated in Figure 1. In general, we force the self-connection matrix $\mathbf{V}$ to be symmetric.

Based on the *spike response model* scheme [12], Eq. (1) has the following solution with the boundary condition $u_{rest} = 0$
$$\boldsymbol{u}_i(t) = \int_{t'}^{t} \exp\left(\frac{s - t'}{\tau_m}\right) \left(\sum_{j \in [n]} \mathbf{V}_{ij}\boldsymbol{s}_j(t) + \sum_{k \in [m]} \mathbf{W}_{ik}\mathbf{I}_k(t)\right) \mathrm{d}s \ , \tag{2}$$

where $t'$ denotes the last firing time $t' = \max\{s \mid \boldsymbol{u}_i(s) = u_{firing}, \ s < t\}$ for a pre-given firing threshold $u_{firing} > 0$. Spiking neuron model employs the typical threshold rule, that is, neuron $i$ fires spikes $\boldsymbol{s}_i(t)$ at time $t$ if and only if $\boldsymbol{u}_i(t) \geq u_{firing}$. After firing, the membrane potential is instantaneously reset to a lower value $u_{rest}$ (rest voltage). Note that this work does not consider using absolute refractory periods [14] or refractory kernels [9].

Let the timing set $\mathbb{T}_i = \{t \mid \boldsymbol{u}_i(t) = u_{firing}, t \in [0, T]\}$ record the firing times of neuron $i$ and $N_i = |\mathbb{T}_i|_\#$. In general, we consider using the *firing rate* $f_i^{\mathrm{ave}}(T) = N_i/T$ to characterize the spike dynamics, where $f_i^{\mathrm{ave}}(T)$ indicates the average number of spikes of neuron $i$ at time interval $[0, T]$. It is clearly observed that $f_i^{\mathrm{ave}}(t)$ is discontinuous since $N_i$ is a step function regarding $\boldsymbol{u}_i(t)$ and time $t$. To ensure the well-posed characteristics of the firing rate functions, we here employ the spike excitation function $f_e : u \to s$ to smooth the firing procedure, for example,

$$
\begin{cases}
\text{Linear:} \quad f_e(\boldsymbol{u}_i(t)) \triangleq \dfrac{\boldsymbol{u}_i(t)}{u_{firing}}, \\[4mm]
\text{Cos-based:} \quad f_e(\boldsymbol{u}_i(t)) \triangleq \left[1 - \cos\left(\dfrac{\pi}{2} \dfrac{t - t^{(k)}}{t^{(k+1)} - t^{(k)}}\right)\right] \left\lfloor \dfrac{\boldsymbol{u}_i(t)}{u_{firing}} \right\rfloor,
\end{cases}
\tag{3}
$$

where $t^{(k)}$ and $t^{(k+1)}$ are two adjacent timings from $\mathbb{T}_i$ in which $k \in \mathbb{N}^+$ and $t^{(k)} < t^{(k+1)}$. Similar smooth treatments on spike excitation functions can refer to [13, 33]. Thus, $f_i^{\mathrm{ave}}(T)$ can be approximated by an **Instantaneous Firing Rate** (IFR) function

$$
f_i(\boldsymbol{x}, t) = f_e(\boldsymbol{u}_i(t))/(t - t'),
\tag{4}
$$

where $\boldsymbol{x} \in \mathbb{R}^m$, $t' \in \mathbb{T}_i$, and $t \in [T]$. It is observed that the proposed IFR function $f_i(\boldsymbol{x}, t)$ induces a discrete dynamical system, in which $f(\cdot, t) : \mathbb{R}^m \to \mathbb{R}$ is a Spatial function and $f(\boldsymbol{x}, \cdot) : \mathbb{R} \to \mathbb{R}$ is a temporal flow.

**About Neural Encoding.** The actual input data (e.g., image or video) should usually be pre-converted into a spiking version before fed up to SNNs. The conversion procedure is called *neural encoding*, as shown in Figure 1. There are two main categories of neural encoding approaches: *temporal encoding* and *rate-based encoding*; the former encodes input data by exploiting the distance between time instances that fire spikes [40], and the latter encodes input data as a count sequence of the fired spikes within temporal windows [18]. The rate-based encoding is the simplest and most popular scheme in SNNs. The representative techniques are usually encoded by a Poisson distribution or recorded by a dynamic vision sensor [3, 27]. Recent years have witnessed a lot of efforts on the information capacity of neural encoding, specially rate-based encoding, from empirical [6, 17] and theoretical [24, 35, 36] sides. Throughout this paper, we adopt rate-based encoding as the default and focus on the firing rates of SNNs, generalizing the computational powers concerning the spike count.

**About Firing Rates.** When we investigate the dynamics and neural computation of SNNs, the *firing rates* or equally the number of firing spikes are the key measure of network activities for investigating neural computation and model dynamics because of the close relation between firing rates and network function (including neural input, connectivity, spiking function, and firing process) [1, 2]. There are great efforts to use firing rates in SNNs for some real-world tasks, such as vision [13, 33, 44] and speech recognition [30, 39]. Besides, Barrett et al. [5] and Chou et al. [8] showed that the averaged firing rate can approximate the optimal solutions of some quadratic programs within polynomial complexity. This work employs an "instantaneous" firing rate rather than the averaged firing rate or the total number of firing spikes used in previous studies. This manner provides a feasible way to construct the discrete dynamical systems using the IFR functions, based on which we can develop in-depth understandings of SNNs from spatial and temporal aspects.

## 4   Approximation to Discrete Dynamical Systems

In this section, we first present the universal approximation for scSNNs, and then show the parameters and time complexities of the IFR functions led by scSNNs for approximating discrete dynamical systems from spatial and temporal aspects, respectively.

### 4.1   Universal Approximation

Now, we present our first theorem about the universal approximation of scSNNs as follows:

**Definition 1** *Let $l \in \mathbb{N}^+$. The function $f(x)$ is said to be **l-finite** if $f$ is an $l$-times differentiable scalar function that satisfies*

$$0 < \left| \int_{\mathbb{R}} D^l f(x) \, \mathrm{d}x \right| < \infty \, .$$

**Theorem 1** *Let $K \subset \mathbb{R}^m$ be a compact set. If the spike excitation function $f_e$ is $l$-finite and $w_i \in \mathbb{R}$ where $i \in [n]$, then for all $r \in [l]$, there exists some time $t$ such that the set of IFR functions $f(\cdot, t) : K \to \mathbb{R}$ of the form $f(\boldsymbol{x}, t) = \sum_{i \in [n]} w_i f_i(\boldsymbol{x}, t)$ is dense in $\mathcal{C}^r(K, \mathbb{R})$.*

Theorem 1 shows that the scSNN is a universal approximator, which provides a solid cornerstone for developing SNNs' theory.

The proof idea of this theorem can be summarized as follows. We utilize the invertibility of the Fourier transform on Sobolev space $\mathcal{W}_\mu^{l,p}(K, \mathbb{R})$ ($p > 1$), to project the concerned functional space $\mathcal{C}^r(K, \mathbb{R})$ into a characteristic space, and the corresponding objective function is transformed as a single integral over the characteristic space. According to Fubini's theorem, the approximation problem on $\mathcal{C}^r(K, \mathbb{R})$ can be converted into another that uses multiple integrals to a single integral on the characteristic space. The subsequent proof can then be completed along the thought lines of the technical exposition given by Carslaw and Rogosinski [7]. The full proof of Theorem 1 can be obtained in Appendix A.

### 4.2 $\epsilon$-Approximation with Parameters Complexity

This subsection presents our second theorem about the parameters complexity of scSNNs as follows:

**Definition 2** *We say that $g$ is a radial function if $g(\boldsymbol{x}') = g(\boldsymbol{x})$ for any $\|\boldsymbol{x}'\| = \|\boldsymbol{x}\|$.*

**Theorem 2** *Given a compact set $K^m \subset \mathbb{R}^m$, a probability measure $\mu$, and a radial function $g : K^m \to \mathbb{R}$. For some apposite spike excitation function $f_e$ and any $\epsilon > 0$, there exists some time $t$ such that the radial function $g$ can be well approximated by a one-hidden-layer scSNN of $\mathcal{O}(Cm^{15/4})$ spiking neurons, that is,*

$$\|f(\boldsymbol{x}, t) - g(\boldsymbol{x})\|_{L^2(\mu)} < \epsilon \, .$$

Theorem 2 shows that the scSNNs with polynomial spiking neurons can well approximate a broad scope of radial functions, which may shed some insights on that scSNNs admit input rotations [36], such as the rotation-based data argument techniques [29] and invariant models [23].

This paper provides two ways for proving this theorem. Here, we only introduce the proof idea of the interesting one (full proof is shown in Appendix B), and another proof way can be accessed in Appendix C. This proof idea can be summarized as follows. The radial function is invariant to rotations and dependent on the input norm, corresponding to the phase and norm (i.e., radial) of inputs, respectively. It is observed that $\boldsymbol{s}(t)$ in Eq. (1) induces a local recurrent function concerning the input $\boldsymbol{x}(t)$. Thus, there exists a collection of parameters such that the IFR function is dependent on the norm of inputs. Besides, the IFR function admits the phase since $\mathbf{V}$ is a symmetric matrix. Thus, there exist some linear connections (including rotation transformations) such that the weighted aggregation of these spiking neurons is invariant to rotations. Summing up the approximation recognition, there is a family of radial functions that can be well approximated by one-hidden-layer scSNNs.

We formally begin our proof of Theorem 2 with some useful lemmas.

**Lemma 1** *Let $g : [r, R] \to \mathbb{R}$ be an $L$-Lipschitz function for $r \leq R$. For any $\delta > 0$, $C_s \geq 1$, and $n \leq C_s R^2 Lm/(\sqrt{r}\delta)$, there exist some time $t$ and an IFR function $f(\boldsymbol{x}, t)$ led by a one-hidden-layer scSNN of $n$ spiking neurons such that*

$$\sup_{\boldsymbol{x} \in \mathbb{R}^m} |g(\|\boldsymbol{x}\|) - f(\boldsymbol{x}, t)| \leq \delta \, .$$

**Lemma 2** *For $m > C_2 > 0$, $g : \mathbb{R}^m \to \mathbb{R}$ is an $L$-Lipschitz radial function supported on the set*

$$\mathcal{S}_\Delta = \{\boldsymbol{x} : 0 < C_2\sqrt{m} \leq \|\boldsymbol{x}\| \leq 2C_2\sqrt{m}\} \, .$$

*For any $\delta > 0$, there exist some time $t$ and an IFR function $f(\boldsymbol{x}, t)$ led by scSNNs of one-hidden layer with width at most $C_s(C_2)^{3/2}L(m)^{7/4}/\delta$ such that*

$$\sup_{\boldsymbol{x} \in \mathbb{R}^m} |g(\boldsymbol{x}) - f(\boldsymbol{x}, t)| < \delta .$$

Lemma 2 shows that the $L$-Lipschitz radial functions can be approximated by the IFR function $f(\boldsymbol{x}, t)$ led by scSNNs of one-hidden layer with polynomial parameters.

**Lemma 3** *Define*

$$g(\boldsymbol{x}) = \sum_{i=1}^{N} \epsilon_i g_i(\boldsymbol{x}) \quad \text{with} \quad g_i(\boldsymbol{x}) = \mathbb{I}\{\|\boldsymbol{x}\| \in \Omega_i\} , \tag{5}$$

*where $\epsilon_i \in \{-1, +1\}$, $N$ is a polynomial function of $m$, and $\Omega_i$'s are disjoint intervals of width $\mathcal{O}(1/N)$ on values in the range $\Theta(\sqrt{m})$. For any $\epsilon_i \in \{-1, +1\}$ ($i \in [N]$), there exists a Lipschitz function $h \colon \mathcal{S}_\Delta \to [-1, +1]$ such that*

$$\int_{\mathbb{R}^m} (g(\boldsymbol{x}) - h(\boldsymbol{x}))^2 \, \phi^2(\boldsymbol{x}) \, \mathrm{d}\boldsymbol{x} \leq \frac{3}{(C_2)^2 \sqrt{m}} .$$

Lemma 3 shows that any non-Lipschitz function $h(\boldsymbol{x})$ can be approximated and bounded by a Lipschitz function with density $\phi^2$.

*Proof of Theorem 2.* Let $g(\boldsymbol{x}) = \sum_{i=1}^{N} \epsilon_i g_i(\boldsymbol{x})$ be defined by Eq. (5) and $N \geq 4C_2^{5/2}m^2$. According to Lemma 3, there exists a Lipschitz function $h$ with range $[-1, +1]$ such that

$$\|h(\boldsymbol{x}) - g(\boldsymbol{x})\|_{L_2(\mu)} \leq \frac{\sqrt{3}}{C_2(m)^{1/4}} .$$

Based on Lemmas 1 and 2, any Lipschitz radial function supported on $\mathcal{S}_\Delta$ can be approximated by an IFR function $f(\boldsymbol{x}, t)$ led by scSNNs of one-hidden layer with width at most $C_3 C_s(m)^{15/4}$, where $C_3$ is a constant relative to $C_2$ and $\delta$. This means that there exists some time $t$ such that

$$\sup_{\boldsymbol{x} \in \mathbb{R}^m} |h(\boldsymbol{x}) - f(\boldsymbol{x}, t)| \leq \delta .$$

Thus, we have

$$\|h(\boldsymbol{x}) - f(\boldsymbol{x}, t)\|_{L_2(\mu)} \leq \delta .$$

Hence, the range of $f(\cdot, t)$ is in $[-1 - \delta, +1 + \delta] \subseteq [-2, +2]$. Provided the radial function, defined by Eq. (5), we have

$$\|g(\boldsymbol{x}) - f(\boldsymbol{x}, t)\|_{L_2(\mu)} \leq \|g(\boldsymbol{x}) - h(\boldsymbol{x})\|_{L_2(\mu)} + \|h(\boldsymbol{x}) - f(\boldsymbol{x}, t)\|_{L_2(\mu)} \leq \frac{\sqrt{3}}{C_2(m)^{1/4}} + \delta .$$

This implies that given constants $m > C_2 > 0$ and $C_3 > 0$, for any $\delta > 0$ and $\epsilon_i \in \{-1, +1\}$ ($i \in [N]$), there exists some time $t$, such that the target radial function $g$ can be approximated by an IFR function $f(\boldsymbol{x}, t)$ led by scSNNs of one-hidden layer with range in $[-2, +2]$ and width at most $C_3 C_s(m)^{15/4}$, that is,

$$\|g(\boldsymbol{x}) - f(\boldsymbol{x}, t)\|_{L_2(\mu)} \leq \frac{\sqrt{3}}{C_2(m)^{1/4}} + \delta < \delta_1 .$$

This completes the proof. $\qquad\square$

### 4.3 $\epsilon$-Approximation with Time Complexity

The IFR function generates a discrete dynamical system, comprising a Spatial function and a temporal flow. Both Theorems 1 and 2 focus on the Spatial (e.g., spatial) approximation characteristics of the IFR function. This subsection investigates its temporal characteristics, especially how long it takes for a self-connected SNN to achieve a specified task or target function. Now, we present our third theorem about the time complexity of scSNNs as follows.

**Definition 3** *A matrix $\mathbf{V} \in \mathbb{R}^{m \times n}$ (if $m \leq n$) is said to be **non-degenerate** if any $m \times m$ sub-matrix of $\mathbf{V}$ has full rank.*

**Theorem 3** *Let $n \geq m$. Let $\boldsymbol{x}_k(t) \in \pi(\lambda_k)$ and $\mathbb{E}(\boldsymbol{x}_k) = \lambda_k$ for $\lambda_k > 0$, $k \in [m]$, and $t \in [T]$. If $\mathbf{V}$ is a non-degenerate matrix, then for any $\epsilon > 0$ and matrix $\mathbf{G} \in \mathbb{R}^{m \times n}$ with $\|\mathbf{G}\|_2 < \infty$, when the time complexity satisfies*

$$T \geq \Omega \left( \frac{\sqrt{n}\, \|\mathbf{G}\|_2}{\epsilon \sqrt{\|\mathbf{V}\|_2}} \right),$$

*there exists some one-hidden-layer scSNNs with the IFR vector $\boldsymbol{f} = (f_1, \ldots, f_n)^\top$ such that*

$$\|\mathbb{E}_{\boldsymbol{x}}\left[\mathbf{V}\boldsymbol{f}(\boldsymbol{x}, T)\right] - \mathbf{G}\boldsymbol{\lambda}\|_2 < \epsilon. \tag{6}$$

Theorem 3 shows that without training, the scSNNs can well approximate the multivariate spike flows (MSF, i.e., linear functions of $\boldsymbol{\lambda}$) within an explicit polynomial time complexity. In contrast to the conventional ANN theory and our proposed theorems above that show the spatial approximation ability, this theorem portrays the computational efficiency of a flow function led by scSNNs. Besides, there are few studies on the computational efficiency of SNNs, and we summarize the comparative results in Table 1. Notice that approximating specific functions, such as MSF in this work, is more challenging than solving some problems with locally competitive algorithms [34]. Thus, it is observed that Theorem 3 provides a rigorous guarantee for SNNs to solve some algorithmic problems. Besides, Theorem 3 relaxes the dimension of the self-connection matrix $\mathbf{V}$ from $n \times n$ to $n \times m$, which provides a more general result for approximating MSF with any dimension.

Table 1: Comparative Results on Computational Efficiency of SNNs.

| Works | Models | Objects | Computational Complexity |
|-------|--------|---------|--------------------------|
| Tang et al. [35] | configured SNNs | solving Sparse Coding Problem | Convergence |
| Chou et al. [8] | simple SNNs | solving Quadratic Programming | Polynomial Complexity |
| Our Work | scSNNs with Poisson inputs | approximating MSF | Polynomial Complexity |

The proof idea of this theorem originates from the attractor theory in dynamical systems. One key lemma is as follows.

**Lemma 4** *Given $\mathbf{U} = \sqrt{\boldsymbol{\Lambda}_U}\ \widetilde{\mathbf{U}}^\top, \tilde{\boldsymbol{\lambda}} \in \mathbb{R}^n$, and $\epsilon > 0$, when $t \geq \Omega\left(\frac{\sqrt{n}\, \|\Lambda_G\|_2}{\epsilon\, \|\mathbf{U}\|_2}\right)$, it holds $\|\mathbf{U}\tilde{\boldsymbol{f}}(\tilde{\boldsymbol{\lambda}}, t) - \tilde{\boldsymbol{\lambda}}\| \leq \epsilon$, where the modified IFR function $\tilde{\boldsymbol{f}}(\tilde{\boldsymbol{\lambda}}, t)$ is led by a scSNN without connection matrix $\mathbf{W}$ and fed up to the constant spike sequence $\tilde{\boldsymbol{\lambda}}$ at every timestamp.*

Lemma 4 shows that when fed up to a constant spike sequence $\tilde{\boldsymbol{\lambda}}$ at every timestamp, a weighted sum of IFR functions can converge to an attractor around the inputs within a polynomial temporal computation. Based on the results of Lemma 4, it suffices to prove that the expectation of the concerned IFR function $\boldsymbol{f}(\boldsymbol{x}, t)$ can approximate $\tilde{\boldsymbol{f}}(\tilde{\boldsymbol{\lambda}}, t^*)$ within time interval $[0, T]$ where $t^* \leq T \leq \Omega(\frac{\sqrt{n}\, \|\mathbf{G}\|_2}{\epsilon \sqrt{\|\mathbf{V}\|_2}})$. Full proof of Theorem 3 can be obtained in Appendix D.

The aforementioned results, time complexity bound in detail, can be verified by a simulated experiment. We here simulate a $4 \times 10,000$ spike sequence from the Iris data sets with a timestamp of $0.001$ using Poisson encoding. We employ the one-hidden-layer scSNN [44] that contains self-connection structure as the conducted SNN model. The number of input channels and hidden neurons are 4 and 10, respectively. The self-connection matrix $\mathbf{V}$ is randomly sampled from $[0, 1]$ with bias $= 1/3$. The above configurations meet the conditions of Theorem 3. Table 2 lists the hyper-parameter values in the conducted scSNN. For any linear matrix $\mathbf{G} \in \mathbb{R}^{4 \times 10}$ with $\|\mathbf{G}\|_2 < \infty$, we define an indicator $t_c = \frac{\sqrt{n}\, \|\mathbf{G}\|_2}{\epsilon \sqrt{\|\mathbf{V}\|_2}}$. Thus, by exploiting the relation among $\epsilon$, $t$, and $t_c$, we can verify the explicit polynomial bound, especially the order of magnitude function $\Omega(\cdot)$ in Theorem 3.

Figure 2 plots the experimental results. From Figure 2(a), the conducted scSNN first approximates the objective function at a faster rate and then maintains a lower approximation error. Figure 2(b) signifies that $\log(\epsilon)$ is inversely proportional to $\log(t_c)$, i.e., $a\log(t_c) + b\log(\epsilon) = c$ where $a, b, c > 0$. Figure 2(c) shows the relation plots between $\log(t_c)$ and $\log(t)$. Notice that the conducted ScNN

Table 2: Hyper-parameter Setting of scSNNs.

| Parameters | Value | Parameters | Value |
|---|---|---|---|
| Time Step | 0.001 | Firing Threshold | 1 |
| Expect Spike Count (True) | 100 | Membrane Time $\tau_m$ | 0.2 |
| Expect Spike Count (False) | 10 | Time Constant of Synapse $\tau_s$ | 0.008 |
| Encoding Length $T$ | 10, 000 | Maximum Firing | 10 |
| Refractory Period | 0.016 | Maximum Time | 10 |

can achieve the same approximation error at different timestamps, we only care about the shortest one, that is, the blue solid curve below the red dashed line. It is observed that $\log(t)$ is proportional to $\log(t_c)$, i.e., $\log(t) = a \log(t_c) + b$ where $a > 0$ and $b \in \mathbb{R}$. In summary, we have demonstrated the effectiveness of our theoretical results.

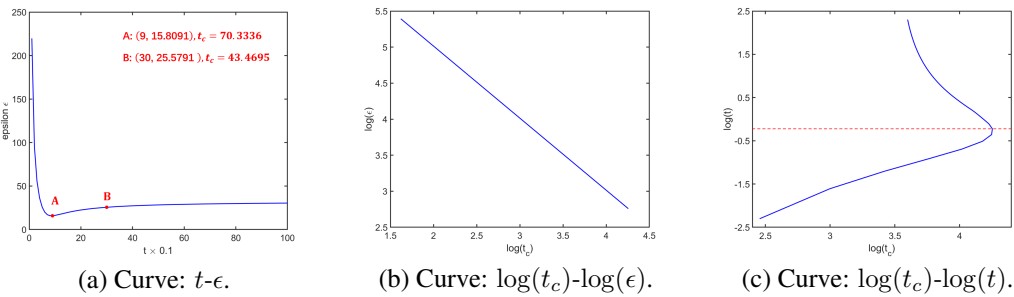

(a) Curve: $t$-$\epsilon$.  (b) Curve: $\log(t_c)$-$\log(\epsilon)$.  (c) Curve: $\log(t_c)$-$\log(t)$.

Figure 2: Magnitude Curves of $\epsilon$, $t$, and $t_c$.

## 5 Discussions

Despite an increasing focus on the potential of handling spatio-temporal data, the theoretical understandings of many aspects (e.g., approximation ability and computational efficiency) of SNNs are still far from clear. Some seminal studies focused on the universality of SNNs. Maass et al. [19, 20, 21] showed that some typical SNNs can simulate the standard computational models such as Turing machines, random access machines, threshold circuits, etc. She et al. [31] showed some universal approximation properties of SNNs by exploiting the spike propagation paths. However, to the best of our knowledge, few theoretical guarantees on the approximation complexity and computational efficiency of SNNs have been provided. There are only some academic studies on the convergence in the limit results for SNNs solving the sparse coding problem [34, 35] and the convergence rates for SNNs solving temporal quadratic programming [8]. Recently, two studies [41] and [44] provided calculable ways for approaching the lower and upper bounds of adaptive SNN systems, respectively.

Our starting point is a recent advance proposed by Zhang and Zhou [44], in which they showed that adding self connections enables SNNs to achieve adaptive dynamical systems with spatio-temporal representation. This work developed an in-depth analysis of approximation ability and computational efficiency of self connections in SNNs. It is observed that self connections facilitate our results; re-using firing spike variables $s(t)$ contributes to approximating the norm of input sequences $x(t)$, and the symmetric self-connection matrix coincides with rotation and dual transformations, which are crucial in the proofs of Theorems 2 and 3. Our theoretical derivatives not only show the universal approximation properties of scSNNs but also disclose the parameters and time complexities for scSNNs approximating discrete dynamical systems, which prospectively provide solid support for the theory development of SNNs. Unfortunately, this paper has not yet shown the theoretical advantages of scSNNs over non-self-connection SNNs (it is valuable to be studied in the future). But we believe that the current results have disclosed the importance and power of self connections for enhancing the approximation and computational ability of SNNs, which may shed some insights on developing provable and sound SNNs.

In light of the preceding merits, many issues are worthy of being studied in the future. One important future issue is to explore some practical techniques for scSNNs. For example, Theorem 2 shows the power of scSNNs on representing radial or equally rotation-invariant functions. So we conjecture that adding self connections may be more compatible with some invariant models and data augmentation techniques, such as image rotation [23, 29]. Besides, the current implementation roughly follows the ideas of Zhang and Zhou [44], i.e., the effect from the $k$-th neuron to the $i$-th neuron equals to the last spike of neuron $k$ multiplied by a weighted factor as shown in Eq. (1). Such a self-connection graph enable SNNs to maintain adaptive characteristics for representing discrete dynamical systems. However, it inevitably leads to a larger memory consumption when the input spike sequences are high-dimensional and high-frequency. Therefore, it is prospective to explore some more practical techniques or modules for scSNNs, which may open up possibilities for achieving sound SNNs.

Another important future issue is to develop in-depth theoretical understandings of SNNs, from aspects of approximation complexity, computational efficiency, representation ability [15, 26, 43], and over-parameterized architectures [4, 45]. Furthermore, it is interesting to explore the theoretical advantages of SNNs with self connections over SNNs without ones, especially from the perspectives of approximation, optimization, and generalization.

## 6   Conclusions

In this paper, we present the theoretical understandings of the approximation ability and computational efficiency of the self-connection SNNs, i.e., scSNNs. We provided three main theorems to show the universal approximation properties, parameters complexity for spatial approximation, and time complexity for temporal approximation of the scSNNs. Our theoretical results disclose the effects of self connections of scSNNs for approximating discrete dynamical systems using polynomial number of parameters within time complexities.

## Acknowledgments and Disclosure of Funding

This research was supported by the National Natural Science Foundation of China (NSFC 61921006) and the Collaborative Innovation Center of Novel Software Technology and Industrialization.

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
