# Supplementary Materials of Theoretically Provable Spiking Neural Networks (Appendix)

In this Appendix, we provide the supplementary materials for our work "Theoretically Provable Spiking Neural Networks", constructed according to the corresponding sections therein. Before that, we review the useful notations as follows.

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

where $\mathcal{B}$ is an abbreviation of the unit ball $\mathcal{B}(1)$. For continuous functions $f, g$, we have the following equalities under Fourier transform

$$\|\widehat{f\phi} - \widehat{g\phi}\|_{L_2(\mu)} = \|f\phi - g\phi\|_{L_2(\mu)} \,,$$

and

$$\widehat{f\phi} = \hat{f} * \hat{\phi}, \quad \text{for} \quad \text{the convolution operator } * \,.$$

# A  Full Proof for Theorem 1

For convenience, we abbreviate $f(\cdot, t)$ as $f(\cdot)$ in this proof. For $r \in [l]$, we have

$$
\begin{aligned}
D^r f(\boldsymbol{x}) &= \int_{\mathbb{R}^m} \widehat{D^r f}(\boldsymbol{y}) \exp\left(2\pi\, \mathrm{i}\boldsymbol{y}^\top \boldsymbol{x}\right) \mathrm{d}\boldsymbol{y} \\
&= \int_{\mathbb{R}^m} \widehat{D^r f}(\beta\boldsymbol{y}) \exp\left(2\pi\beta\,\mathrm{i}\boldsymbol{y}^\top \boldsymbol{x}\right) \mathrm{d}(\beta\boldsymbol{y}) \\
&= \int_{\mathbb{R}^m} (2\pi\beta\,\mathrm{i}\boldsymbol{y})^r \widehat{f}(\beta\boldsymbol{y}) \exp\left(2\pi\beta\,\mathrm{i}\boldsymbol{y}^\top \boldsymbol{x}\right) |\beta|^m \, \mathrm{d}\boldsymbol{y} \\
&= \int_{\mathbb{R}^m} \left[\boldsymbol{y}^r |\beta|^m \widehat{f}(\beta\boldsymbol{y})\right] \left[(2\pi\beta\,\mathrm{i})^r \exp\left(2\pi\beta\,\mathrm{i}\boldsymbol{y}^\top \boldsymbol{x}\right)\right] \mathrm{d}\boldsymbol{y} \\
&= \int_{\mathbb{R}^m} \frac{\boldsymbol{y}^r |\beta|^m \widehat{f}(\beta\boldsymbol{y})}{\widehat{f}_e(\beta)} \left[\widehat{D^r f_e}(\beta) \exp\left(2\pi\beta\,\mathrm{i}\boldsymbol{y}^\top \boldsymbol{x}\right)\right] \mathrm{d}\boldsymbol{y} \\
&= \int_{\mathbb{R}^m} \frac{\boldsymbol{y}^r |\beta|^m \widehat{f}(\beta\boldsymbol{y})}{\widehat{f}_e(\beta)} \left[\int_{\mathbb{R}} D^r f_e(\alpha) \exp\left(-2\pi\,\mathrm{i}\beta\alpha\right) \mathrm{d}\alpha\right] \exp\left(2\pi\beta\,\mathrm{i}\boldsymbol{y}^\top \boldsymbol{x}\right) \mathrm{d}\boldsymbol{y} \,,
\end{aligned}
\tag{7}
$$

where $\alpha, \beta \in \mathbb{R}$, and the above equations hold from the Fourier transforms and some of their properties. By taking the real part of Eq. (7), we have

$$
D^r f(\boldsymbol{x}) = \int_{\mathbb{R}^m} \int_{\mathbb{R}} \boldsymbol{y}^r D^r f_e(\alpha) \mathcal{K}(\alpha, \beta, \boldsymbol{y}) \, \mathrm{d}\alpha \, \mathrm{d}\boldsymbol{y} \,,
\tag{8}
$$

where

$$
\mathcal{K}(\alpha, \beta, \boldsymbol{y}) = \frac{|\beta|^m \widehat{f}(\beta\boldsymbol{y}) \exp\left[2\pi\beta\,\mathrm{i}(\boldsymbol{y}^\top \boldsymbol{x} - \alpha)\right]}{\widehat{f}_e(\beta)} \,.
$$

In this proof, we set

$$
\alpha = \boldsymbol{y}^\top \boldsymbol{x} + \hbar, \quad \boldsymbol{y} = \mathbf{W}_{i,[m]}, \quad \hbar = \sum_{j \in [n]} -\frac{1}{\tau_m} \exp\left(-\frac{s - t'}{\tau_m}\right) \mathbf{V}_{i,j} \boldsymbol{s}_j(t') \,,
$$

and the $k$-th element of vector $\boldsymbol{x}$ equals to a temporal-weighted average of $\mathbf{I}_k(t)$ at time interval $[t', t]$

$$
\boldsymbol{x}_k = \int_{t'}^{t} \exp\left(-\frac{s - t'}{\tau_m}\right) \mathbf{I}_k(s) \, \mathrm{d}s.
$$

Thus, we have

$$
\mathcal{K}(\alpha, \beta, \boldsymbol{y}) = \frac{|\beta|^m \widehat{f}(\beta\boldsymbol{y}) \exp\left(2\pi\beta\hbar\,\mathrm{i}\right)}{\widehat{f}_e(\beta)} \triangleq \mathcal{K}_\beta(\hbar, \boldsymbol{y}) \quad \text{and} \quad \sup_{\boldsymbol{x} \in K} |\boldsymbol{x}| \leq C_x \,.
$$

Based on Eq. (8), we can construct a family of approximation functions of the form

$$
f_\kappa(\boldsymbol{x}) = \int_{\mathcal{B}_1} \int_{\mathcal{B}_2} f_e(\boldsymbol{y}^\top \boldsymbol{x} + \hbar) \mathcal{K}_\beta(\hbar, \boldsymbol{y}) \, \mathrm{d}\hbar \, \mathrm{d}\boldsymbol{y} \,,
\tag{9}
$$

where $\mathcal{B}_1 = \{\boldsymbol{x} \mid \boldsymbol{x} \preccurlyeq \kappa\}$ and $\mathcal{B}_2 = \{\boldsymbol{x} \mid \boldsymbol{x} \preccurlyeq (C_x m + 1)\kappa\}$. Thus, we have

$$
D^r f_\kappa(\boldsymbol{x}) = \int_{\mathcal{B}_1} \int_{\mathcal{B}_2} \boldsymbol{y}^r D^r f_e(\boldsymbol{y}^\top \boldsymbol{x} + \hbar) \mathcal{K}_\beta(\hbar, \boldsymbol{y}) \, \mathrm{d}\hbar \, \mathrm{d}\boldsymbol{y} \,.
\tag{10}
$$

It suffices to prove that $D^r f_\kappa \to D^r f$ uniformly on $K$, as $\kappa \to \infty$. Now

$$
\begin{aligned}
D^r f_\kappa(\boldsymbol{x}) - D^r f(\boldsymbol{x}) &= \int_{\mathbb{R}^m / \mathcal{B}_1} \int_{\mathcal{R}} \boldsymbol{y}^r D^r f_e(\boldsymbol{y}^\top \boldsymbol{x} + \hbar) \mathcal{K}_\beta(\hbar, \boldsymbol{y}) \, \mathrm{d}\hbar \, \mathrm{d}\boldsymbol{y} \\
&\quad + \int_{\mathcal{B}_1} \int_{\mathbb{R} / \mathcal{B}_2} \boldsymbol{y}^r D^r f_e(\boldsymbol{y}^\top \boldsymbol{x} + \hbar) \mathcal{K}_\beta(\hbar, \boldsymbol{y}) \, \mathrm{d}\hbar \, \mathrm{d}\boldsymbol{y} \\
&\triangleq \mathcal{R}_1 + \mathcal{R}_2 \,.
\end{aligned}
$$

For $R_1$, one has

$$\begin{aligned}
|\mathcal{R}_1| &= \left| \int_{\mathbb{R}^m/\mathcal{B}_1} \int_{\mathcal{R}} \boldsymbol{y}^r D^r f_e(\boldsymbol{y}^\top \boldsymbol{x} + \hbar) \mathcal{K}_\beta(\hbar, \boldsymbol{y}) \, \mathrm{d}\hbar \, \mathrm{d}\boldsymbol{y} \right| \\
&\leq \int_{\mathbb{R}^m/\mathcal{B}_1} |\boldsymbol{y}^r| \left| \int_{\mathcal{R}} D^r f_e(\boldsymbol{y}^\top \boldsymbol{x} + \hbar) \mathcal{K}_\beta(\hbar, \boldsymbol{y}) \, \mathrm{d}\hbar \right| \mathrm{d}\boldsymbol{y} \\
&\leq \int_{\mathbb{R}^m/\mathcal{B}_1} |\boldsymbol{y}^r| \left| \int_{\mathcal{R}} D^r f_e(\boldsymbol{y}^\top \boldsymbol{x} + \hbar) \, \mathrm{d}\hbar \right| \left| \frac{|\beta|^m \widehat{f}(\beta \boldsymbol{y})}{\widehat{f}_e(\beta)} \right| \mathrm{d}\boldsymbol{y} \\
&\leq \left\| D^r f_e(\boldsymbol{y}^\top \boldsymbol{x} + \hbar) \right\|_{1, \mathbb{R}} \int_{\mathbb{R}^m/\mathcal{B}_1} \left| \frac{|\beta|^m \boldsymbol{y}^r \widehat{f}(\beta \boldsymbol{y})}{\widehat{f}_e(\beta)} \right| \mathrm{d}\boldsymbol{y} \\
&\leq \left\| D^r f_e(\boldsymbol{y}^\top \boldsymbol{x} + \hbar) \right\|_{1, \mathbb{R}} \int_{\mathbb{R}/\tilde{\mathcal{B}}_1} \left| \frac{|\beta \boldsymbol{y}|^r \widehat{f}(\beta \boldsymbol{y})}{\widehat{f}_e(\beta)|\beta|^r} \right| \mathrm{d}(\beta \boldsymbol{y}) \\
&\leq \frac{\left\| D^r f_e(\boldsymbol{y}^\top \boldsymbol{x} + \hbar) \right\|_{1, \mathbb{R}}}{\left| \widehat{f}_e(\beta)|\beta|^r \right|} \int_{\mathbb{R}/\tilde{\mathcal{B}}_1} \left| \boldsymbol{y}^r \widehat{f}(\boldsymbol{y}) \right| \mathrm{d}\boldsymbol{y} ,
\end{aligned}$$

where $\tilde{\mathcal{B}}_1 = \{\beta \boldsymbol{x} \mid \beta \boldsymbol{x} \preccurlyeq \beta \kappa\}$. For $R_2$, one has

$$\begin{aligned}
|\mathcal{R}_2| &= \left| \int_{\mathcal{B}_1} \int_{\mathbb{R}/\mathcal{B}_2} \boldsymbol{y}^r D^r f_e(\boldsymbol{y}^\top \boldsymbol{x} + \hbar) \mathcal{K}_\beta(\hbar, \boldsymbol{y}) \, \mathrm{d}\hbar \, \mathrm{d}\boldsymbol{y} \right| \\
&\leq \int_{\mathcal{B}_1} \left| \int_{\mathbb{R}/\mathcal{B}_2} D^r f_e(\boldsymbol{y}^\top \boldsymbol{x} + \hbar) \, \mathrm{d}\hbar \right| \left| \frac{|\beta|^m \boldsymbol{y}^r \widehat{f}(\beta \boldsymbol{y})}{\widehat{f}_e(\beta)} \right| \mathrm{d}\boldsymbol{y} \\
&\leq \int_{\mathbb{R}/\tilde{\mathcal{B}}_2} |D^r f_e(\mu)| \, \mathrm{d}\mu \cdot \int_{\tilde{\mathcal{B}}_1} \left| \frac{|\beta \boldsymbol{y}|^r \widehat{f}(\beta \boldsymbol{y})}{\widehat{f}_e(\beta)|\beta|^r} \right| \mathrm{d}(\beta \boldsymbol{y}) \\
&\leq \int_{\mathbb{R}/\tilde{\mathcal{B}}_2} |D^r f_e(\mu)| \, \mathrm{d}\mu \, \frac{\|D^r f_e(\mu)\|_{1, \tilde{\mathcal{B}}_1}}{\left| \widehat{f}_e(\beta)|\beta|^r \right|} ,
\end{aligned}$$

where $\mu = \boldsymbol{y}^\top \boldsymbol{x} + \hbar$ and $\tilde{\mathcal{B}}_2 = \{x \mid x \preccurlyeq \kappa\}$ since $|\mu| \geq \kappa$. Summing up the inequalities above, we have

$$\sup_{\boldsymbol{x} \in K} |D^r f_\kappa(\boldsymbol{x}) - D^r f(\boldsymbol{x})| \leq \frac{C_\kappa^1 + C_\kappa^2}{\left| \widehat{f}_e(\beta)|\beta|^r \right|}$$

with

$$C_\kappa^1 = \left\| D^r f_e(\boldsymbol{y}^\top \boldsymbol{x} + \hbar) \right\|_{1, \mathbb{R}} \int_{\mathbb{R}/\tilde{\mathcal{B}}_1} \left| \boldsymbol{y}^r \widehat{f}(\boldsymbol{y}) \right| \mathrm{d}\boldsymbol{y} \quad \text{and} \quad C_\kappa^2 = \left\| D^r f_e(\mu) \right\|_{1, \tilde{\mathcal{B}}_1} \int_{\mathbb{R}/\tilde{\mathcal{B}}_2} |D^r f_e(\mu)| \, \mathrm{d}\mu ,$$

which tends to 0 as $\kappa \to \infty$.

Given $\kappa$, it suffices to construct a series of approximations to $f_\kappa$ in Eq. (9). Formally, we define

$$\tilde{f}_\kappa^n(\boldsymbol{x}) = \sum_{\boldsymbol{\mu} \in \mathcal{U}} \tilde{\beta} f_e(\tilde{\boldsymbol{y}}^\top \boldsymbol{x} + \tilde{\hbar}) ,$$

where

$$\begin{cases}
\boldsymbol{\mu} = (\mu_1, \mu_2, \ldots, \mu_m)^\top \quad \text{with} \quad \mu_i \in [-n, n] \cap \mathbb{Z} \quad \text{for} \quad i \in [m], \\
\tilde{\beta} = (C_x m + 1)(\kappa/n)^{m+1} \mathcal{K}_\beta(\tilde{\hbar}, \tilde{\boldsymbol{y}}) , \\
\tilde{\boldsymbol{y}} = \boldsymbol{\mu} \kappa/n , \\
\tilde{\hbar} = \mu^*(C_x m + 1)\kappa/n \quad \text{with} \quad \mu^* \in [-n, n] \cap \mathbb{Z} .
\end{cases}$$

It is observed that $\tilde{f}_\kappa^n$ belongs to the set of IFR functions, and

$$D^r \tilde{f}_\kappa^n(\boldsymbol{x}) = \sum_{\boldsymbol{\mu} \in \mathcal{U}} (C_x m + 1)(\kappa/n)^{m+1} \tilde{\boldsymbol{y}}^r D^r f_e(\tilde{\boldsymbol{y}}^\top \boldsymbol{x} + \tilde{\hbar}) \mathcal{K}_\beta(\tilde{\hbar}, \tilde{\boldsymbol{y}}) . \tag{11}$$

Next, we are going to prove that $D^r \tilde{f}_\kappa^n \to D^r f_\kappa$ uniformly on $K$, as $n \to \infty$. For simplicity, we define the following function

$$G_\beta(\boldsymbol{x}, \boldsymbol{y}, \hbar) = \boldsymbol{y}^r D^r f_e(\boldsymbol{y}^\top \boldsymbol{x} + \hbar) \mathcal{K}_\beta(\hbar, \boldsymbol{y}) \ .$$

Thus, Eq. (10) and Eq. (11) become

$$D^r f_\kappa(\boldsymbol{x}) = \sum_{\boldsymbol{\mu} \in \mathcal{U}} \int_{\mathcal{B}_3} G_\beta(\boldsymbol{x}, \boldsymbol{y}, \hbar) \, \mathrm{d}\hbar \, \mathrm{d}\boldsymbol{y}$$

and

$$D^r \tilde{f}_\kappa^n(\boldsymbol{x}) = \sum_{\boldsymbol{\mu} \in \mathcal{U}} \int_{\mathcal{B}_3} G_\beta(\boldsymbol{x}, \tilde{\boldsymbol{y}}, \tilde{\hbar}) \, \mathrm{d}\hbar \, \mathrm{d}\boldsymbol{y}$$

respectively, where $\cup_{\boldsymbol{\mu} \in \mathcal{U}} \mathcal{B}_3 = \{(x_0, x_1, \ldots, x_m) \mid x_0 \in \mathcal{B}_2, \ (x_1, \ldots, x_m)^\top \in \mathcal{B}_1\} \subset \mathbb{R}^{m+1}$. Hence, one has

$$\sup_{(\hbar, \boldsymbol{y}), (\tilde{\hbar}, \tilde{\boldsymbol{y}}) \in \mathcal{B}_3} \left| G_\beta(\boldsymbol{x}, \boldsymbol{y}, \hbar) - G_\beta(\boldsymbol{x}, \tilde{\boldsymbol{y}}, \tilde{\hbar}) \right| < \infty \ .$$

Let

$$C_\kappa^n(\delta) \quad \triangleq \quad \sup_{\substack{(\hbar, \boldsymbol{y}), (\tilde{\hbar}, \tilde{\boldsymbol{y}}) \in \mathcal{B}_3 \\ |(\hbar, \boldsymbol{y}) - (\tilde{\hbar}, \tilde{\boldsymbol{y}})| \le \delta^{m+1}}} \left| G_\beta(\boldsymbol{x}, \boldsymbol{y}, \hbar) - G_\beta(\boldsymbol{x}, \tilde{\boldsymbol{y}}, \tilde{\hbar}) \right| \ .$$

Thus, we have

$$\left| D^r \tilde{f}_\kappa^n(\boldsymbol{x}) - D^r f_\kappa(\boldsymbol{x}) \right| \le \sum_{\boldsymbol{\mu} \in \mathcal{U}} \int_{\mathcal{B}_3} \left| G_\beta(\boldsymbol{x}, \boldsymbol{y}, \hbar) - G_\beta(\boldsymbol{x}, \tilde{\boldsymbol{y}}, \tilde{\hbar}) \right| \mathrm{d}\hbar \, \mathrm{d}\boldsymbol{y}$$

$$\le \sum_{\boldsymbol{\mu} \in \mathcal{U}} \int_{\mathcal{B}_3} C_\kappa^n(\kappa/n) \, \mathrm{d}\hbar \, \mathrm{d}\boldsymbol{y}$$

$$\le C_\kappa^n(\kappa/n) \sum_{\boldsymbol{\mu} \in \mathcal{U}} \int_{\mathcal{B}_3} \mathrm{d}\hbar \, \mathrm{d}\boldsymbol{y}$$

$$\le C_\kappa^n(\kappa/n) \, (2n)^{m+1} \, (C_x m + 1)(\kappa/n)^{m+1},$$

where the last inequality holds from

$$\int_{\mathcal{B}_3} \mathrm{d}\hbar \, \mathrm{d}\boldsymbol{y} = (C_x m + 1)(\kappa/n)^{m+1} \quad \text{and} \quad |\mathcal{U}|_\# = (2n)^{m+1}.$$

Further, we can obtain

$$\sup_{\boldsymbol{x} \in K} \left| D^r \tilde{f}_\kappa^n(\boldsymbol{x}) - D^r f_\kappa(\boldsymbol{x}) \right| \le (C_x m + 1)(2\kappa)^{m+1} \, C_\kappa^n(\kappa/n) \ ,$$

which tends to 0 as $n \to \infty$.

Finally, we finish the proof by taking double limits $n \to \infty$ before $\kappa \to \infty$. $\qquad \square$

# B   Full Proof for Theorem 2

There are two proof methods for Theorem 2, one based on the rotation group action and one based on the Fourier transformation, which are detailed in this and the following sections, respectively.

This proof idea can be summarized as follows. It is observed that the IFR function comprises the radius and phase since $\mathbf{V}$ is a symmetric matrix. Thus, there exist some linear connections (including rotation transformations) such that the combination of these spiking neurons is invariant to rotations. In other words, the IFR function led by scSNNs with one-hidden layer can easily and well approximate some radial functions, see Lemma 1, since the radial function is invariant to rotations, and is dependent on the input norm.

## B.1   Proof of Lemma 1

Let $f' : \mathbb{R}^m \to \mathbb{R}$ be a radial function with $f'(\boldsymbol{x}) = \|\boldsymbol{x}\|$. According to Theorem 1, for any $\delta > 0$, $i \in [n]$, and $m \geq 1$, there exists some time $t$ such that

$$\sup_{\boldsymbol{x} \in \mathbb{R}^d} \left| f'(\boldsymbol{x}) - |f_i(\boldsymbol{x}, t)| \right| \leq \delta/2 \, , \tag{12}$$

where $f_i$ denotes the IFR function of the $i$-th hidden spiking neuron, defined by Eq. (4). Further, we define a new function $g' : \mathbb{R} \to \mathbb{R}$ as follows.

$$g'(s) = \sum_{i=1}^{n'} \alpha_i' f_e(s) \, ,$$

where $\alpha_i', a_i' \in \mathbb{R}$. For Lipschitz continuous function $\sqrt[r]{\cdot}$ and from [42, Lemma 1], we have

$$\sup_{s \in [r^k, R^k]} \left| g(\sqrt[k]{s}) - g'(s) \right| \leq \delta/2 \, , \tag{13}$$

where $n' \leq C'L(R^k - r^k)/(\sqrt[k]{r}\delta)$ for some constant $C' > 0$ and integer $k \geq 2$. Further, we have

$$|g'(s) - f(\boldsymbol{x}, t)| \leq |g'(s) - f'(\boldsymbol{x})| + |f'(\boldsymbol{x}) - f(\boldsymbol{x}, t)| \, , \tag{14}$$

where

$$f'(\boldsymbol{x}) = \sum_{i=1}^{n'} w_i' \, |f_i(\boldsymbol{x}, t))| \, ,$$

in which $\{w_i'\}$ denotes another collection of weighted parameters that corresponds to $f(\cdot, t) = \sum_{i \in [n]} w_i f_i(\cdot, t)$. The first term of Eq. (14) can be bounded by $\delta/4$ from [42, Lemma 1] for any $s \in [r^k, R^k]$. The second term is at most $\delta/4$ when $n \geq n'$ from Eq. (12). This follows that

$$|g'(s) - f(\boldsymbol{x}, t)| \leq \delta/2 \, . \tag{15}$$

Combining with Eqs. (13) and (15), we have

$$|g(\|\boldsymbol{x}\|) - f(\boldsymbol{x}, t)| \leq \left| g(\sqrt[k]{s}) - g'(s) \right| + |g'(s) - f(\boldsymbol{x}, t)| \leq \delta \, ,$$

where $\boldsymbol{x} \in \mathbb{R}^m$ and $s \in [r^k, R^k]$. We finally obtain

$$n \leq C_s(R^k - r^k)Lm/(\sqrt[k]{r}\delta) \, ,$$

provided $n \leq mn'$ and $C' \leq C_s$. Finally, we can complete the proof by setting $k = 2$ in the above upper bound. $\qquad\square$

## B.2   Proof of Lemma 2

Let $r = C_2\sqrt{m}$, $R = 2C_2\sqrt{m}$, and $m \geq 1$, then we have $r \geq 1$, which satisfies the condition of [42, Lemma 1]. Invoke Lemma 1 to construct the concerned spiking neural networks and define $\delta' \leq \delta/d$. Then for any $L$-Lipschitz radial function $g : \mathbb{R}^m \to \mathbb{R}$ supported on $\mathcal{S}_\Delta$, we have

$$\sup_{\boldsymbol{x} \in \mathbb{R}^m} |g(\boldsymbol{x}) - f(\boldsymbol{x}, t)| \leq \delta' \, ,$$

where the width of the hidden layer is bounded by

$$n \leq \frac{C_s(C_2)^{3/2}mL}{\delta}(m)^{3/4} \leq \frac{C_s(C_2)^{3/2}L}{\delta}(m)^{7/4} \, .$$

This completes the proof. $\qquad\square$

## B.3 Proof of Lemma 3

Define a branch function

$$h_i(\boldsymbol{x}) = \begin{cases} \max\{\mathbb{I}\{\|\boldsymbol{x}\| \in \Omega_i\}, ND_i\}, & \text{if} \quad B_i = 1, \\ \boldsymbol{0} & , \quad \text{if} \quad B_i = 0, \end{cases}$$

with

$$D_i = \min\left\{\left|\|\boldsymbol{x}\| - \left(1 + \frac{i-1}{N}\right)C_2\sqrt{m}\right|, \left|\|\boldsymbol{x}\| - \left(1 + \frac{i}{N}\right)C_2\sqrt{m}\right|\right\}.$$

Let

$$h(\boldsymbol{x}) = \sum_{i=1}^{N} \epsilon_i h_i(\boldsymbol{x}),$$

$B_i = 1$, $\Omega_i$'s are disjoint intervals, $h_i(\boldsymbol{x})$ is an $N$-Lipschitz function. Thus, $h$ is also an $N$-Lipschitz function. So we have

$$\int_{\mathbb{R}^m} \left(h(\boldsymbol{x}) - \sum_{i=1}^{N} \epsilon_i g_i(\boldsymbol{x})\right)^2 \phi^2(\boldsymbol{x})\,\mathrm{d}\boldsymbol{x} = \int_{\mathbb{R}^m} \sum_{i=1}^{N} \epsilon_i^2 \left(h_i(\boldsymbol{x}) - g_i(\boldsymbol{x})\right)^2 \phi^2(\boldsymbol{x})\,\mathrm{d}\boldsymbol{x}$$

$$= \sum_{i=1}^{N} \int_{\mathbb{R}^m} \left(h_i(\boldsymbol{x}) - g_i(\boldsymbol{x})\right)^2 \phi^2(\boldsymbol{x})\,\mathrm{d}\boldsymbol{x}$$

$$\leq (3/(C_2)^2\sqrt{m}),$$

where the last inequality holds from [10, Lemma 22]. This completes the proof. □

Based on the lemmas above, we can finish the proof of Theorem 2.

*Finishing the Proof of Theorem 2.* Let $g(\boldsymbol{x}) = \sum_{i=1}^{N} \epsilon_i g_i(\boldsymbol{x})$ be defined by Eq. (5) and $N \geq 4C_2^{5/2}m^2$. According to Lemma 3, there exists a Lipschitz function $h$ with range $[-1, +1]$ such that

$$\|h(\boldsymbol{x}) - g(\boldsymbol{x})\|_{L_2(\mu)} \leq \frac{\sqrt{3}}{C_2(m)^{1/4}}.$$

Based on Lemmas 1 and 2, any Lipschitz radial function supported on $\mathcal{S}_\Delta$ can be approximated by an IFR function $f(\boldsymbol{x}, t)$ led by scSNNs of one-hidden layer with width at most $C_3 C_s(m)^{15/4}$, where $C_3$ is a constant relative to $C_2$ and $\delta$. This means that there exists some time $t$ such that

$$\sup_{\boldsymbol{x} \in \mathbb{R}^m} |h(\boldsymbol{x}) - f(\boldsymbol{x}, t)| \leq \delta.$$

Thus, we have

$$\|h(\boldsymbol{x}) - f(\boldsymbol{x}, t)\|_{L_2(\mu)} \leq \delta.$$

Hence, the range of $f(\cdot, t)$ is in $[-1 - \delta, +1 + \delta] \subseteq [-2, +2]$. Provided the radial function, defined by Eq. (5), we have

$$\|g(\boldsymbol{x}) - f(\boldsymbol{x}, t)\|_{L_2(\mu)} \leq \|g(\boldsymbol{x}) - h(\boldsymbol{x})\|_{L_2(\mu)} + \|h(\boldsymbol{x}) - f(\boldsymbol{x}, t)\|_{L_2(\mu)} \leq \frac{\sqrt{3}}{C_2(m)^{1/4}} + \delta.$$

This implies that given constants $m > C_2 > 0$ and $C_3 > 0$, for any $\delta > 0$ and $\epsilon_i \in \{-1, +1\}$ ($i \in [N]$), there exists some time $t$, such that the target radial function $g$ can be approximated by an IFR function $f(\boldsymbol{x}, t)$ led by scSNNs of one-hidden layer with range in $[-2, +2]$ and width at most $C_3 C_s(m)^{15/4}$, that is,

$$\|g(\boldsymbol{x}) - f(\boldsymbol{x}, t)\|_{L_2(\mu)} \leq \frac{\sqrt{3}}{C_2(m)^{1/4}} + \delta < \delta_1.$$

This completes the proof. □

# C   Another Proof for Theorem 2

The another proof idea can be summarized as follows. Given any permutation operation on the input channels, there are some permutation operations on the self-connection and final connection weights such that the output of SNNs are invariant. In other words, the IFR function led by scSNNs with one-hidden layer admits the rotation transformations. Therefore, it suffices to show that before final weighted aggregation, the component IFR functions of hidden spiking neurons can approximate any $L$-Lipschitz radial function, similar to the proof line of Lemma 1, and then find a special radial function that can be well approximated by these IFR functions within the polynomial parameter complexity, similar to the proof line of Lemma 3.

**Lemma 5** *Let $f_e$ and $f(\boldsymbol{x}, t)$ be an l-finite function and the IFR function led by scSNNs with one-hidden layer, respectively. For any $\delta > 0$ and $\mathbf{A} \in \mathbf{SO}(m, \mathbb{R})$, there exists some time $t$ such that*

$$|f(\mathbf{A}\boldsymbol{x}, t) - \|\boldsymbol{x}\|| < \delta .$$

**Lemma 6** *Let $f_e$ be an l-finite function, $g : \mathbb{R} \to \mathbb{R}$ is a scalar function, and $f(\boldsymbol{x}, t)$ is the IFR function led by scSNNs with one-hidden layer, where $\boldsymbol{x} \in \mathcal{S}(r)$ and $\mathcal{S}(r)$ is a sphere supported with density $\phi^2$ for $0 < r < \infty$. For any $\delta > 0$, if there exists some time $t \in \mathbb{R}$ such that it holds*

$$|f(\boldsymbol{x}, t) - g(\|\boldsymbol{x}\|)| < \delta \quad \text{if and only if} \quad |f(\mathbf{A}\boldsymbol{x}, t) - g(\|\boldsymbol{x}\|)| < \delta ,$$

*for any $\mathbf{A} \in \mathbf{SO}(m, \mathbb{R})$.*

**Lemma 7** *Define a radial function $g' : \mathbb{R} \to \mathbb{R}$ with the form of*

$$g'(\|\boldsymbol{x}\|) \stackrel{\text{def}}{=} \sum_{i=1}^{N} \epsilon_i \, \mathbb{I}\{\|\boldsymbol{x}\| \in \Omega_i\} ,$$

*where $N$ is a polynomial function of $m$, $\boldsymbol{\epsilon} = (\epsilon_1, \ldots, \epsilon_N) \in \{-1, +1\}^N$, and $\Omega_i$'s are disjoint intervals of width $\mathcal{O}(1/N)$ on values in the range $\Theta(\sqrt{m})$. For $C_2, C_3 > 0$ with $d > C_2$, any $\delta > 0$, and any choice of $\epsilon_i \in \{-1, +1\}$ ($i \in [N]$), there exist some time $t$ and $\mathbf{A} \in \mathbf{SO}(m, \mathbb{R})$ such that*

$$|g'(\|\boldsymbol{x}\|) - f(\mathbf{A}\boldsymbol{x}, t)| \leq \frac{\sqrt{3}}{C_2(m)^{1/4}} + \delta ,$$

*where $f(\boldsymbol{x}, t)$ indicates the IFR function led by scSNNs of one-hidden layer with range in $[-2, +2]$ and at most $C_3 C_s(m)^{15/4}$ hidden spiking neurons.*

*Finishing the Proof of Theorem 2.*    The rest proof is a straightforward combination of Lemmas 5, 6 and 7. Define a radial function $g' : \mathbb{R} \to \mathbb{R}$ with the form of

$$g'(\|\boldsymbol{x}\|) \stackrel{\text{def}}{=} \sum_{i=1}^{N} \epsilon_i \, \mathbb{I}\{\|\boldsymbol{x}\| \in \Omega_i\} ,$$

where $N$ is a polynomial function of $m$, $\boldsymbol{\epsilon} = (\epsilon_1, \ldots, \epsilon_N) \in \{-1, +1\}^N$, and $\Omega_i$'s are disjoint intervals of width $\mathcal{O}(1/N)$ on values in the range $\Theta(\sqrt{m})$. Obviously, $g'$ is equivalent to another radial function defined in Eq. (5) as follows

$$g(\boldsymbol{x}) = \sum_{i=1}^{N} \epsilon_i g_i(\boldsymbol{x}) \quad \text{with} \quad g_i(\boldsymbol{x}) = \mathbb{I}\{\|\boldsymbol{x}\| \in \Omega_i\} ,$$

provided $\boldsymbol{x} \in \mathcal{S}(r)$. From Lemma 7, it is observed that provided the concerned radial function $g(\boldsymbol{x})$ with a collection of choices $\boldsymbol{\epsilon}$, there exist some time $t$ and $n \in \mathcal{O}(C_3 C_s(m)^{15/4})$ such that $g(\boldsymbol{x})$ can be approximated a IFR function $f(\boldsymbol{x}, t)$ led by one-hidden-layer scSNNs of with range in $[-2, +2]$ and at most $n$ hidden spiking neurons, such that

$$|g'(\|\boldsymbol{x}\|) - f(\mathbf{A}\boldsymbol{x}, t)| \leq \frac{\sqrt{3}}{C_2(m)^{1/4}} + \delta .$$

Let $\mathbf{A} \in \mathbf{SO}(m, \mathbb{R})$. If $\|\boldsymbol{x}\| \in \Omega_i \subseteq \mathcal{S}(r)$ for $0 < r < \infty$, then it holds $\|\mathbf{A}\boldsymbol{x}\| \in \Omega_i \subseteq \mathcal{S}(r)$ and $g'(\|\boldsymbol{x}\|) = g'(\|\mathbf{A}\boldsymbol{x}\|)$. According to Lemma 6, we have

$$\|g(\boldsymbol{x}) - f(\boldsymbol{x}, t)\|_{L_2(\mu)} = |g'(\|\boldsymbol{x}\|) - f(\boldsymbol{x}, t)| = |g'(\|\boldsymbol{x}\|) - f(\mathbf{A}\boldsymbol{x}, t)| \leq \frac{\sqrt{3}}{C_2(m)^{1/4}} + \delta ,$$

without any incremental change in the parameter complexity. This completes the proof.    $\square$

# D  Full Proof for Theorem 3

We begin this proof with an investigation that acting SVD on the matrix $\mathbf{G}$. Let $\mathbf{G} = \mathbf{P}\,\mathbf{\Lambda}_G\,\mathbf{Q}^\top$, where $\mathbf{P} \in \mathbb{R}^{m \times m}$ and $\mathbf{Q} \in \mathbb{R}^{n \times n}$ are two unitary matrices, $\mathbf{\Lambda}_G \in \mathbb{R}^{m \times n}$ is a diagonal matrix. Thus, we have

$$\left\| \mathbb{E}_{\boldsymbol{x}}\left[ \mathbf{V}\boldsymbol{f}(\boldsymbol{x},t) \right] - \mathbf{G}\boldsymbol{\lambda} \right\|_2 = \left\| \mathbf{Q}\widetilde{\mathbf{V}}\,\mathbb{E}_{\boldsymbol{x}}\left[ \boldsymbol{f}(\boldsymbol{x},t) \right] - \mathbf{P}\mathbf{\Lambda}_G\boldsymbol{\lambda} \right\|_2 \left\| \mathbf{Q}^\top \right\|_2 \leq 2 \left\| \widetilde{\mathbf{V}}\,\mathbb{E}_{\boldsymbol{x}}\left[ \boldsymbol{f}(\boldsymbol{x},t) \right] - \mathbf{\Lambda}_G\boldsymbol{\lambda} \right\|_2 . \tag{16}$$

If the matrix $\mathbf{G}$ is degenerate, then $\mathbf{\Lambda}_G$ contains some null diagonal elements. Then there must exists a non-degenerate sub-matrix $\mathbf{G}_{\text{sub}}$ of $\mathbf{G}$ such that Inequality (6) degenerates to one about $\mathbf{G}_{\text{sub}}$ in a lower dimensional space within at most constant time cost. Thus, we only care about the case that $\mathbf{G}$ is a non-degenerate matrix. When $m \leq n$, the vector $\mathbf{\Lambda}_G\boldsymbol{\lambda}$ becomes a constant vector, denoted as $\tilde{\boldsymbol{\lambda}}$, whose elements consist of $\lambda_1, \ldots, \lambda_m$ and $n - m$ zeros.

Notice that the $n \times n$-dimensional matrix $\widetilde{\mathbf{V}}$ is symmetric and non-degenerate, since $\mathbf{V}$ is a symmetric and non-degenerate matrix and $\mathbf{Q} \in \mathbf{O}(n)$. Thus, we have $\widetilde{\mathbf{V}} = \widetilde{\mathbf{U}}\,\mathbf{\Lambda}_U\,\widetilde{\mathbf{U}}^\top$, where $\widetilde{\mathbf{U}} \in \mathbf{O}(n)$ and $\mathbf{\Lambda}_U = \text{diag}\{\rho_1, \ldots, \rho_n\}$ in which $\rho_k \neq 0$ for any $k \in [n]$. Then we have the following lemma.

**Lemma 8 (Lemma 4 as aforementioned)** *Given* $\mathbf{U} = \sqrt{\mathbf{\Lambda}_U}\,\widetilde{\mathbf{U}}^\top, \tilde{\boldsymbol{\lambda}} \in \mathbb{R}^n$, *and* $\epsilon > 0$, *when* $t \geq \Omega\left( \frac{\sqrt{n}\,\|\mathbf{\Lambda}_G\|_2}{\epsilon\,\|\mathbf{U}\|_2} \right)$, *it holds* $\|\mathbf{U}\tilde{\boldsymbol{f}}(\tilde{\boldsymbol{\lambda}}, t) - \tilde{\boldsymbol{\lambda}}\| \leq \epsilon$, *where the modified IFR function* $\tilde{\boldsymbol{f}}(\tilde{\boldsymbol{\lambda}}, t)$ *is led by a scSNN without connection matrix* $\mathbf{W}$ *and fed up to the constant spike sequence* $\tilde{\boldsymbol{\lambda}}$ *at every timestamp.*

This lemma is a straightforward derivation of [8, Theorem 1] due to the following two facts. (1) For any $0 < \gamma \leq \min\{\rho_k, k \in [n]\}$, the matrix $\mathbf{U}$ meets the regular conditions of [8, Definition 1]. (2) The optimal solution of $\mathbf{U}\boldsymbol{z} = \tilde{\boldsymbol{\lambda}}$ becomes $\boldsymbol{z}^* = \mathbf{U}^{-1}\tilde{\boldsymbol{\lambda}}$. Hence, we omit the detailed proof of this lemma.

Finally, it suffices to prove that the concerned IFR function $\boldsymbol{f}(\boldsymbol{x}, t)$ can approximate $\tilde{\boldsymbol{f}}(\tilde{\boldsymbol{\lambda}}, t^*)$ within time interval $[0, T]$ where $t^* \leq T$ and $T \geq \Omega(\frac{\sqrt{n}\,\|\mathbf{G}\|_2}{\epsilon\sqrt{\|\mathbf{V}\|_2}})$. Since $f_e$ is the linear function defined by Eq. (3). The approximation above can be converted into another one between

$$\boldsymbol{u}_i(t) = \int_{t'}^t \exp\left( \frac{s - t'}{\tau_m} \right) \left( \sum_{j \in [n]} \mathbf{V}_{ij}\boldsymbol{s}_j(t) + \sum_{k \in [m]} \mathbf{W}_{ik}\boldsymbol{x}_k(t) \right) \mathrm{d}s$$

and

$$\tilde{\boldsymbol{u}}_i(t^*) = \int_{t''}^{t^*} \exp\left( \frac{s - t'}{\tau_m} \right) \tilde{\boldsymbol{\lambda}}_i(s)\,\mathrm{d}s .$$

Since

$$\mathbb{E}\left[ \boldsymbol{u}_i(t) \right] = \int_{t'}^t \exp\left( \frac{s - t'}{\tau_m} \right) \left( \sum_{j \in [n]} \mathbf{V}_{ij}\mathbb{E}\left[ \boldsymbol{s}_j(t) \right] + \sum_{k \in [m]} \mathbf{W}_{ik}\mathbb{E}\left[ \boldsymbol{x}_k(t) \right] \right) \mathrm{d}s$$

$$= \int_{t'}^t \exp\left( \frac{s - t'}{\tau_m} \right) \left( \sum_{j \in [n]} \mathbf{V}_{ij}\mathbb{E}\left[ \boldsymbol{s}_j(t) \right] + \sum_{k \in [m]} \mathbf{W}_{ik}\lambda_k(t) \right) \mathrm{d}s ,$$

we have

$$\left\| \mathbb{E}\left[ \boldsymbol{u}(T) \right] - \tilde{\boldsymbol{u}}(t^*) \right\| \leq \epsilon, \quad \left\| \widetilde{\mathbf{U}}\,\mathbb{E}\left[ \boldsymbol{u}(T) \right] - \tilde{\boldsymbol{u}}(t^*) \right\| \leq \epsilon, \quad \text{and} \quad \left\| \mathbf{U}\,\mathbb{E}\left[ \boldsymbol{u}(T) \right] - \tilde{\boldsymbol{u}}(t^*) \right\| \leq \epsilon .$$

Thus, we have

$$\left\| \mathbf{U}\,\mathbb{E}\left[ \boldsymbol{f}(\boldsymbol{x}, T) \right] - \tilde{\boldsymbol{f}}(\boldsymbol{x}, t^*) \right\| \leq \epsilon .$$

We finally complete the proof according to $\|\mathbf{G}\|_2 = \|\mathbf{\Lambda}_G\|_2$ and $\sqrt{\|\mathbf{V}\|_2} = \|\mathbf{U}\|_2$. $\qquad\square$