# OpenReview forum: "Theoretically Provable Spiking Neural Networks"
_NeurIPS.cc/2022/Conference — NeurIPS 2022 Accept_

### Official Review · Reviewer_6Qhf · 2022-07-09

**Rating:** 6
**Confidence:** 3
**Soundness:** 3 good
**Presentation:** 3 good
**Contribution:** 3 good

**Summary:**

This manuscript, theoretically studies the approximation powers and computational efficiency of spiking neural networks with self-connections (scSNNs)). It is demonstrated that the self-connection structure enables spiking neural networks to approximate continuous dynamical systems within polynomial parameters and time complexities.

**Questions:**

1- Why Theorems 1 \& 2 only consider compact subsets $K \subseteq \mathbb{R}^m$? If the assumption of compactness is removed or at least weakened, can both theorems be still proven?

**Limitations:**

The authors have discussed the limitations of the work and there is no potential negative societal impact.



**Strengths And Weaknesses:**

**Strengths**

- The paper is ideally well supported by theoretical arguments. Some proofs (given in both the main text and Appendix) are written in a nice and technical way. Providing summarized proof ideas after theorems is very helpful.

- The paper is well organized and clearly written. The problem was well motivated and the writing was consistent in tackling this problem throughout the work.


**Weaknesses**
- The paper needs more experimental support/evidence. It would be worthwhile to see at least one more example to evaluate the obtained theoretical results.

*Minor issues*:

-- Fig. 2 is not so visible and needs to be improved. Font size and and axes labels should perhaps be larger.

-- It might be better to number some equations (in the proofs) so that one can refer to them.

-- There are some typos/grammatical errors: Line 525, denote ---> denotes; line 530, we an ---> we can; line 563, "**There**, it is sufficient to show [...]"? Lines 568 \& 570, is ---> be

---

> ### Author Response · Authors · 2022-07-28
> **Response to Reviewer 6Qhf**
>
> We want to thank reviewer 6Qhf for insightful comments.
>
> ----
>
> [1] About "weakening the consideration of compact subsets."
>
> R: A compact set is a broader concept than a bounded set. Informally, it is sufficient and significant to use compact sets to implement approximation theory in conventional neural network theory. In our experience, we may have to consider the distribution of the concerned data for the case that weakens the compact sets. Our work also attempts in Theorem 2, such as introducing a probability measure $\mu$ and its corresponding density function $\phi$. Such proof is more difficult, and we will continue it in future work.
>
> ----
>
> [2] About "improving Fig.2", "numbering some equations", and "some typos/grammatical errors."
>
> R: Thanks again for your carefull suggestions. We got it, and will fix this problem in the revised paper.

---

> > ### Comment · Reviewer_6Qhf · 2022-08-07
> > **Thanks!**
> >
> > Thank you for your response!

---

### Official Review · Reviewer_CgzY · 2022-07-12

**Rating:** 6
**Confidence:** 3
**Soundness:** 3 good
**Presentation:** 2 fair
**Contribution:** 2 fair

**Summary:**

In this work the authors explore spiking neural networks (SNN) with self-connections and their ability to approximate functions spatially and temporally and how computationally efficient they are. The authors show in the paper that  SNNs can approximate approximate continuous dynamical systems within polynomial parameters and time complexities.


**Questions:**

I have no questions for the authors.

**Limitations:**

yes.

**Strengths And Weaknesses:**

The paper builds on previous work showing that self connections are needed in SNN to work on spatiotemporal data due to their bifurcation dynamics. The paper explores the theoretical properties of SNNs with self-connection which is a novel contribution. However, since it closely follows the previous paper, which already showed that self-connections are needed. the originality is very limited. The theoretical approach used in this paper is also not novel. The theory presented in the paper is sound and there are no major errors as far as I can understand. The paper is fairly clearly presented. However, I would suggest to improve figure 2 by making the axis labels more legible, they are way too small.  Regarding the significance of the paper, the paper essentially corroborates previous literature and therefore I think its significance is also limited. It is unlikely to make a huge impact, but the work done here was necessary and it has been solidly executed.

---

> ### Author Response · Authors · 2022-07-28
> **Response to Reviewer CgzY**
>
> We thank the reviewer for appreciating all aspects of the work positively. We will make Fig.2 clearer in the revised paper.

---

### Official Review · Reviewer_qm3g · 2022-07-13

**Rating:** 6
**Confidence:** 1
**Soundness:** 3 good
**Presentation:** 2 fair
**Contribution:** 3 good

**Summary:**

The authors prove 3 theorems for "self-connection" (which I understand as recurrent) spiking neural networks (scSNNs), showing that scSNNs are universal approximators, they can approximate radial functions using polynomial spiking neurons, and they can approximate multivariate spike flows within polynomial time, which they also verify via simulations.

**Questions:**

The authors should also connect self-connection SNNs with recurrent SNNs -- to my understanding these are the same and the latter term is far more prevalent.

Rest of the questions / comments are listed above. Overall the manuscript is very dense and has not been connected well to other results and concepts/practices in the field. No effort is made to make the material useful/accessible to a wider audience. If the authors can clarify my points above and provide more intuitive understanding as well, I would be willing to improve my rating.

**Strengths And Weaknesses:**

Looking at equation (1), "self-connection" seems to mean recurrent connections. There are already theorems showing that recurrent neural networks can approximate dynamical systems (e.g. Schäfer, A. M., & Zimmermann, H. G. (2006). Recurrent Neural Networks Are Universal Approximators. https://doi.org/10.1007/11840817_66 or Seidl, D. R., & Lorenz, R. D. (1991). A structure by which a recurrent neural network can approximate a nonlinear dynamic system. https://doi.org/10.1109/IJCNN.1991.155422). The novelty to my understanding here is to show the same with recurrent *spiking* neural networks -- I am unsure if this has been shown already or is new. The authors are advised to clarify what is new compared to the non-spiking (or possibly existing spiking) proofs.

The proof uses spike response model (SRM) for the neurons. Would the proof also go through if a Leaky Integrate and Fire (LIF) model is used for neurons?

The definition of $f(\mathbf{x},t)$ in Theorem 1 seems to be a weighted sum of the instantaneous firing rates of the individual neurons as defined in equation 4. If I understand correctly, theorem 3 seems to suggest that without training, one can decode these instantaneous firing rates to approximate any MSF. This is reminescent of liquid state machine (reservoir computing). Perhaps the authors can highlight what is different and/or connections if any to the same? Also in theorem 3, $\mathbf{V}$ is an nxm matrix -- I suppose this is different from the $\mathbf{V}$ defined in equation 1 as an nxn self-connection matrix. If yes, then please use a different symbol here, else please clarify.

Disclaimer: I am not qualified to comments on the mathematical soundness, and would leave it to other reviewers.

---

> ### Author Response · Authors · 2022-07-28
> **Response to Reviewer qm3g**
>
> We want to thank Reviewer qm3g for the insightful comments.
>
> ----
>
> [1] About "recurrent or self-connection."
>
> R: We confirm it is a self-connection structure, which Reviewer CgzY also admits. In contrast to the conventional SNNs, such as the LIF model of the form
>
> $\frac{d \boldsymbol{u}_i(t)}{d t} = -\frac{1}{\tau_m} \boldsymbol{u}_i(t) + \sum_{k\in[m]} \mathbf{W}_{ik} \mathbf{I}_k(t) $,
>
> the proposed scSNN adopts the idea of a recent advance [Bifurcation Spiking Neural Network. JMLR'2021.] that employs self-connection of the form
>
> $\sum_{j\in[n]} \mathbf{V}_{ij} \boldsymbol{s}_j(t) $.
>
> As illustrated in Fig. 1, our work adds some self-connection lines (blue lines) based on conventional recurrent connections (orange lines).
>
> The difference between our proposed scSNN and BSNN is the different implementations of self-connection. BSNN connects the real-time membrane potentials with linear weights, whereas our work employs the firing signals, reducing computational consumption. Besides, there also are some efforts on alternative implementation, e.g., [Structural Stability of Spiking Neural Networks. arXiv: 2207.04876. 2022.] attempts a higher-order self-connection function.
>
> Informally, we find that once there is no self-connection, even with this recurrence structure, it is not easy to prove the approximation ability of SNNs to the continuous-time dynamical system, not to mention the approximation complexity and computational efficiency.
> Our finding corroborates previous literature.
>
> Thus, an efficient and theory-solid implementation of self-connection is one of the contributions of this paper. Of course, we will add a detailed description of the self-connection structure in Appendix.
>
> ----
>
> [2] About " Would the proof also go through if a Leaky Integrate and Fire (LIF) model is used for neurons?"
>
> R: We have not yet asserted whether the LIF model passes this proof, i.e., no strict proof yet. As a rule of thumb, we believe it is challenging to obtain the same results only using the LIF model. We encounter more significant difficulties, such as the denseness of the IFR function, when directly proving with the LIF model than with the SRM model.
>
> ----
>
> [3] About "Perhaps the authors can highlight what is different and/or connections if any to the same between scSNN and reservoir computing."
>
> R: Indeed, the reservoir computing could also approximate the continuous-time dynamical system, referring to [Reservoir Computing with Dynamical Systems. Ph.D. Thesis of the University of Bath. 2019.] And there are also some theoretical results of reservoir computing. However, there is a considerable difference between the working mechanism of reservoir computing and SNN. For example, the input data fed to SNN is spike sequences, whereas reservoir computing does not have this limitation. Besides, SNNs have the characteristics of low-cost computing due to the mechanism of delayed emission, and thus they have been widely welcomed by high-performance computing and neural computing in recent years. Finally, we will add a discussion to clarify the practical and theoretical comparison among the reservoir computing, recurrent networks, and our proposed scSNN.
>
> ----
>
> [4] About "n \times m" or "n \times n self-connection matrix."
>
> R: Indeed, we force the self-connection matrix to be square throughout this paper, whereas Theorem 3 considers a more general case where one can decode the instantaneous firing rate functions to approximate any MSF with \textbf{any dimension}. Theorem 3 breaks the square property of self-connection matrices. However, it is worth noting that the conclusion of Theorem 3 still holds for square self-connection matrices. Finally, we'll fix it in a future release.
>
> ----
>
> [5] About "connected well to other results."
>
> R: Thanks to the reviewer for the valuable comments. Indeed, we should make some connections (different or the same) with previous results in the broader field, which will increase the work's readability and impact. We promise that we will continue to revise this paper in subsequent versions.
>
> It is worth emphasizing that this work contribures to the theoretical development of SNNs, related to some work such as [Bifurcation Spiking Neural Network. JMLR'2021.] and [On the algorithmic power of spiking neural networks. ITCS'2019]. We have to admit that theoretical work does cause some inconvenience to read. In addition, there is still relatively little academic exploration work on SNNs, as introduced in Section 5 Discussions and Prospects.
>
> Overall, we believe our work corroborates previous literature and was significant for the development of SNNs' theory.

---

> > ### Comment · Reviewer_qm3g · 2022-08-09
> > **Response**
> >
> > Thanks for your reponse.
> > Indeed the self-connection aspect is now clearer, though recurrence often (indeed unless explicitly excluded) includes self-connections.
> > There are both spiking and non-spiking versions of reservoir computing, termed liquid state machines (from the Maass group) and echo state networks (from Jaeger group).
> > Glad that you will bring out the conections further. All the best!

---

### Meta-Review · Area_Chair_LNe4 · 2022-08-24

**Recommendation:** Accept
**Confidence:** Certain

**Metareview:**

This paper shows that spiking neural networks with self-connections can approximate continuous dynamical systems with polynomial parameter and time complexities. All reviewers agreed that the contributions of this paper were clearly above the acceptance threshold.

**Award:**

No

---

### Decision · Program_Chairs · 2022-09-14

Accept